# The Cryogenic Electron Microscopy Structure of the Cell Adhesion Regulator Metavinculin Reveals an Isoform-Specific Kinked Helix in Its Cytoskeleton Binding Domain

**DOI:** 10.3390/ijms22020645

**Published:** 2021-01-11

**Authors:** Erumbi S. Rangarajan, Tina Izard

**Affiliations:** Cell Adhesion Laboratory, Department of Integrative Structural and Computational Biology, The Scripps Research Institute, Jupiter, FL 33458, USA; rerumbi@scripps.edu

**Keywords:** actin, cadherin, cancer, catenin, cell adhesion, cell junction, cell migration, cell signaling, heart failure, integrin, plasma membrane

## Abstract

Vinculin and its heart-specific splice variant metavinculin are key regulators of cell adhesion processes. These membrane-bound cytoskeletal proteins regulate the cell shape by binding to several other proteins at cell–cell and cell–matrix junctions. Vinculin and metavinculin link integrin adhesion molecules to the filamentous actin network. Loss of both proteins prevents cell adhesion and cell spreading and reduces the formation of stress fibers, focal adhesions, or lamellipodia extensions. The binding of talin at cell–matrix junctions or of α-catenin at cell–cell junctions activates vinculin and metavinculin by releasing their autoinhibitory head–tail interaction. Once activated, vinculin and metavinculin bind F-actin via their five-helix bundle tail domains. Unlike vinculin, metavinculin has a 68-amino-acid insertion before the second α-helix of this five-helix F-actin–binding domain. Here, we present the full-length cryogenic electron microscopy structure of metavinculin that captures the dynamics of its individual domains and unveiled a hallmark structural feature, namely a kinked isoform-specific α-helix in its F-actin-binding domain. Our identified conformational landscape of metavinculin suggests a structural priming mechanism that is consistent with the cell adhesion functions of metavinculin in response to mechanical and cellular cues. Our findings expand our understanding of metavinculin function in the heart with implications for the etiologies of cardiomyopathies.

## 1. Introduction

Vinculin and its larger and muscle-specific spliced isoform called metavinculin play pivotal roles in cell adhesion by regulating cellular morphology, cellular motility, and by the transducing force between neighboring cells as well as between cells and the extracellular matrix [1,2,3,4,5,6,7] At cell–cell and cell–matrix junctions, (meta)vinculin functions as a scaffold where it connects transmembrane cell surface receptors to the filamentous actin cytoskeleton [8,9,10].

Vinculin is essential for the development of the heart as well as cardiomyocyte adhesion functions including contraction [11,12]. *Vinculin* knockout mouse embryos have neural tube and cardiac developmental defects and do not survive past embryonic day 10. *Vinculin*-null murine embryonic fibroblasts have fewer adhesions and thus have defects in cell spreading and are less able to attach. Therefore, they are more motile and resist apoptosis and anoikis [13]. Moreover, heterozygous *vinculin* deletion (+/-) results in dilated cardiomyopathies [14].

Vinculin and metavinculin have seven four-helix bundle domains that are connected *via* a about 40 residue (in vinculin) or about 100 residue (in metavinculin) flexible proline-rich linker to their respective and distinct five-helix bundle tail domains, which is called Vt in vinculin, for vinculin tail, or MVt in metavinculin, for metavinculin tail (Figure 1) [15,16,17,18]. Metavinculin has 68 amino acids inserted into the vinculin polypeptide chain between vinculin residues 915 and 916. Structurally, these additional 68 residues replace the N-terminal extended coil and the first α-helix H1 of the five-helix Vt bundle domain, whereby the Vt α-helix H1 in MVt becomes part of the head–tail disordered linker in metavinculin instead. The Vt and MVt domains bind to F-actin, and their primary and tertiary structural differences correlate with their distinct actin bundling and binding properties [19,20,21,22,23].

Vinculin and metavinculin are exclusively helical, which provides these proteins with a large degree of flexibility to easily act as adaptor proteins. Two conformations have been recognized as functionally relevant: (i) their closed and inactive and (ii) their open and active states. Vinculin and metavinculin are held in their closed auto-inhibited conformers by an extensive nanomolar affinity interface between their N-terminal four-helix bundle Vh1 subdomains and their respective Vt or MVt tail domains. In turn, the vinculin and metavinculin binding partners often need to be activated first prior to binding to vinculin and metavinculin. Such activation often occurs by mechanical actomyosin force [4,24,25,26,27,28].

Apart from talin and actin, over a dozen other proteins also bind to vinculin and metavinculin [29]. The binding sites that have been characterized are grouped as binders to (i) the N-terminal Vh1 bundles, as seen for α-actinin [10,30,31,32], α-catenin [33,34,35,36], and β-catenin [37,38]; or (ii) with the proline-rich linker region, as suggested for vinexin α/β [39,40], vasodilator-stimulated phosphoprotein [41,42,43], and the seven-subunit actin related proteins-2/3 [44,45]; or (iii) with the tail domains Vt and MVt, respectively, such as reported for paxillin [46,47] and raver1 [48,49,50]. Pathogens such as *Shigella* and *Rickettsia* bind the N-terminal Vh1 bundle by mimicry to exploit this vinculin binding site for entry into the host cell [51,52,53,54].

Metavinculin is also interesting from a cardiac perspective, since metavinculin is the vinculin isoform that is expressed in both smooth [55] and cardiac muscle cells [56,57] as well as in platelets [58]. In muscle cells, metavinculin expression is enhanced during contraction [59,60]. In mice, heterozygous inactivation or the knockout of *vinculin* results in dilated cardiomyopathy [14,61,62], and metavinculin deficiency leads to disorganized intercalated discs in human patients [63,64]. Metavinculin point mutations are particular to patients with dilated cardiomyopathy [65] as well as hypertrophic cardiomyopathy [66,67]. However, family linkage analyses have not been reported, and the number of identified patients with metavinculin mutations seems too small as direct evidence for metavinculin dysfunction causing cardiomyopathies. Loss of *metavinculin* did not affect the development of the heart and its function [68]. Nevertheless, the vinculin isoforms display distinct mechanical properties, and it will be interesting to determine how metavinculin modulates cell adhesion mechanics.

To further our understanding of metavinculin activation and function, we determined the 4.17 Å cryogenic electron microscopy structure of metavinculin in its native state. Our structure shows that metavinculin captures individual states of this dynamic protein and reveals that metavinculin is much more flexible than anticipated from our crystal structures, whereby its individual domains display a large degree of rotational freedom within the restrictions provided by the tight head–tail auto-inhibition that keeps metavinculin locked in its inactive state. Our results pave the way to further our understanding of metavinculin functions in the heart and its role in devastating cardiomyopathies.

## 2. Results

### 2.1. The Metavinculin Polypeptide Chains in the Crystal Structures are Distinct

Our human wild-type and patient-derived 954 deletion mutant metavinculin crystal structures crystallized with two polypeptide chains in the asymmetric unit that are very similar [16]. The two wild-type subunits (12,257 atoms) can be superimposed onto the Δ954 mutant with root means squares deviations of 0.4 Å (Figure 1B). In the asymmetric unit, the N-terminal four-helix bundle of Vh1 of subunit A packs against the C-terminal four-helix bundle of Vh1 of subunit B as well as the N-terminal four-helix bundle of Vh3. In addition, the tail domain MVt from subunit A packs against the N-terminal four-helix bundle of Vh2.

Subunits B are slightly more ordered in both wild-type and Δ954 mutant metavinculin structures, as is also evident from the additional residues 835–859 of the proline-rich region that are visible in the electron density maps for subunits B that are missing for subunits A. However, subunits A and B show different conformations whereby subunit A superimposes onto subunit B with root means squares deviations of 1.3 Å (for 5969 atoms) (Figure 1C) in the 3.4 Å metavinculin Δ954 mutant structure as for the 3.6 Å metavinculin wild-type structure (not shown).

Comparison of the two polypeptide chains shows that the biologically relevant Vh1–MVt interface is fairly conserved with polypeptide chain movements of under 2 Å (Figure 1C). Perhaps the largest relative domain movements of about 12 Å (as measured at the Cα position of for example residue 456) are seen in the C-terminal four-helix bundle of Vh2 that is almost completely (about 89% or its surface) solvent exposed. Large relative four-helix bundle movements of about 7 Å in the polypeptide chain (as measured at the Cα position of for example residue 341) are also seen in the N-terminal four-helix bundle of this Vh2 domain. Its movement affects the N-terminal four-helix bundle domain of Vh3, where it interacts with where the polypeptide chains have moved by about 4 Å (as measured at the Cα position of for example residue 532). The far end of this N-terminal four-helix bundle domain of Vh3 is again largely solvent exposed (about 72% or its surface), and there, the relative domain movements are larger, about 7 Å as measured at the Cα position of for example residue 562.

Finally, the five-helix Vt2 bundle that is also fairly solvent exposed (about 83% of its surface) shows variations of about 4 Å (as measured at the Cα position of, for example, residue 809). This conformational variability is lower (about 3 Å as measured at the Cα position of for example Cα residue 730), where the Vt2 four-helix bundle movement is restricted by interdomain contacts with the MVt five-helix bundle.

While metavinculin crystallized in space group *P* 4_2_,2_1_2 with two molecules in the asymmetric unit, the monoclinic human 2.9 Å vinculin structure (PDB entry 1tr2) [17] has two subunits in its asymmetric unit that pack much more tightly as documented by their buried surface area of about 1187 Å^2^ and dimensions of the two polypeptide chains are about 130 Å × 127 Å × 128 Å compared to the intermolecular contacts seen in metavinculin (about 964 Å^2^ and 158 Å × 110 Å × 80 Å). This tighter packing of vinculin *versus* metavinculin correlates with the better resolution obtained for vinculin compared to metavinculin. Perhaps the most notable intra-molecular inter-domain differences for any of these four combinations of superpositions are seen in the C-terminal Vh2 four-helix bundle that shows varying degrees of interactions with the N-terminal four-helix Vh1 bundle (Figure 1C).

The 3.1 Å chicken gizzard vinculin (PDB entry 1st6) [18] has one polypeptide chain in the asymmetric unit (space group *C* 222_1_) and superimposes onto the human vinculin subunit A with root means squares deviations of 1.7 Å for 5926 atoms or subunit B with root means squares deviations of 2 Å for 6295 atoms. In this comparison also, the most notable relative domain shifts are seen for the C-terminal Vh2 four-helix bundle that shows varying degrees of interactions with the N-terminal four-helix Vh1 bundle (not shown).

Human vinculin subunit A resembles human metavinculin subunit A best (root means squares deviations of 1.5 Å for 6235 atoms), and human vinculin subunit B superimposes onto human metavinculin subunits A and B similarly (root means squares deviations of 1.9 Å in both cases for 6232 or 6411 atoms, respectively). The most notable intra-molecular inter-domain differences for any of these four combinations of superpositions here also remain for the C-terminal Vh2 four-helix bundle that shows varying degrees of interactions with the N-terminal four-helix Vh1 bundle (not shown).

### 2.2. The Cryogenic Electron Microscopy Metavinculin Structure

To better understand the physiological relevance and biological function of the metavinculin domain flexibility seen in our crystal structures, we determined the cryogenic electron microscopy structure of human metavinculin. The initial processing of a single ab initio three-dimensional reconstruction, including all particles accounting for best two-dimensional classes (Figure 2A), and subsequent homogeneous refinement yielded a 4.17 Å resolution structure (Table 1, Figure 2B) that overall resembled our earlier crystal structures confirming overall the auto-inhibited conformation of metavinculin. The local resolution estimate suggested that the central core is well ordered to 4.2 to 4.5 Å and contributes toward the overall resolution of the metavinculin structure. The peripheral α-helices are in the 5 to 6 Å resolution range, while a few of the surface regions exhibit resolution of less than 8 Å (Figure 2C).

To account for the conformational flexibility within various domains, three-dimensional classification of the initial *ab initio* model was carried out through heterogeneous refinement in cryoSPARC [69] into three main classes. The individual three main classes accounted to about 35%, 32%, and 28% of the total particles available with the remaining 5% segregated as non-aligned particles. Furthermore, these individual three-dimensional classes were subjected to homogeneous refinement to yield 4.15 Å, 4.5 Å, and 4.27 Å conformers representing at least two distinct populations (Figure 3). When comparing all three cryogenic electron microscopy conformers presented here to our crystal structures, greater structural similarity is seen with subunit A in the crystal. The two similar (4.15 Å and 4.27 Å) cryogenic electron microscopy structures superimpose onto subunit A with root means squares deviations of 2.1 and 2.4 Å for 6,359 or 6,320 atoms, respectively versus the distinct 4.5 Å cryogenic electron microscopy structure (3.4 Å for 6468 atoms). Subunit B of the crystal structure superimposes more poorly with superposition values ranging from 3 to 3.6 Å.

### 2.3. The Cryogenic Electron Microscopy Metavinculin Conformers

The nanomolar interaction is the anchor of the auto-inhibited metavinculin (and vinculin) conformer. While this interface is restricting the remaining domains roughly in their places, our cryogenic electron microscopy structure shows that they are surprisingly mobile. When comparing all metavinculin conformations, the biologically relevant head–tail interface that keeps metavinculin (and vinculin) in their autoinhibited conformation seems unaltered (Figure 4A). Indeed, the superposition of Vh1 residues 2–250 and MVt residues 975–1114 from the various metavinculin cryogenic electron microscopy conformations or crystal structures results in root means squares deviations of less than 1 to about 2 Å for 2500–2700 atoms and even when superimposing onto the equivalent vinculin Vh1-Vt domains, such values are up to 2.4 Å for 2749 atoms (Figure 4A).

### 2.4. A Novel Conformer in the F-Actin Binding Domain

Our 4.5 Å resolution cryogenic electron microscopy metavinculin conformer has a novel feature not encountered before. Its first MVt α-helix H1’ is distinctly different from the conformation seen in any other metavinculin structure. In all other full-length structures, α-helix H1’ is parallel to α-helices H3 and H5 or antiparallel to α-helices H2 and H4 of the five-helix bundle MVt domain. Therefore, we name this metavinculin conformation the “H1’-parallel metavinculin” structure. However, in its new conformer, the N-terminus (residues 961–972) of α-helix H1’ are by about 60° rotated from its MVt helix bundle bound state (Figure 4A) to instead interact with the solvent exposed end of the N-terminal four-helix bundle of Vh3 (Figure 4B). We will name this distinct metavinculin conformation the “H1’-kinked metavinculin” structure. Resulting new MVt–Vh3 interactions are provided by MVt residues Gln-964, Gln-971, and Ser-972 and Vh3 residues Gln-501, Asp-505, Glu-565, and Arg-570.

The effects of the N-terminal four-helix bundle of Vh3 that is now closer to MVt in the H1’-kinked metavinculin structure compared to its position in the other metavinculin structures is transferred to the C-terminal Vh3 four-helix bundle that in turn is further away from its interaction seen with the C-terminal four-helix bundle subdomain of Vh1 (Figure 4C). For example, Cα positions of residues 605, 620, 684, and 690 are 8–13 Å further away from their positions (and the Vh1 C-terminal bundle) in the H1’-kinked metavinculin structure compared to the H1’-parallel metavinculin structures.

### 2.5. The Metavinculin Vh2–Vh3 Domain Constellations

The new proximity of the N-terminal four-helix bundle of Vh3 with MVt are also affecting the interaction of Vh3 with Vh2 resulting in a relative shift of the N-terminal four-helix bundle of Vh2 not to lose its interaction with the Vh3 domain (Figure 4B). For example, the N-terminal Vh2 four-helix bundle shows relative movements of its 4 α-helices of up to 9 Å as measured at Cα position of residues 341. Since the N-terminal Vh2 and Vh3 four-helix bundles both moved in the same direction, their respective Vh2–Vh3 distances remain at about 15 Å, as measured at Cα position between residues 298 and 636, in both the H1’-kinked and H1’-parallel metavinculin conformers. This collective relative movement of four-helix bundles is also somewhat the case for the C-terminal four-helix bundle of Vh3 and its relative position to the Vt2 domain (Figure 4C), where the distances, as measured at the Cα position between residues 604 and 747, are about 14 Å and about 17 Å for the H1’-kinked and H1’-parallel metavinculin structures, respectively.

### 2.6. The Two H1’-Parallel Metavinculin Conformations

Our two H1’-parallel cryogenic electron microscopy conformers are quite similar and superimpose with root means squares deviations of less than 1.2 Å for 6614 atoms compared to their poorer superposition with the H1’-kinked metavinculin structure, where these values are 2.9 Å or 2.6 Å for 7097 or 7057 atoms, respectively (not shown). There is a relative shift in the position of the C-terminal (or N-terminal) Vh2 four-helix bundle subdomain of about 4.5 Å (or 5.3 Å) as measured between the Cα positions of residues 449 (or 288) in the H1’-kinked and the H1’-parallel metavinculin structures (Figure 5A). As mentioned above, the H1’-parallel metavinculin cryogenic electron microscopy structures resemble the subunit A crystal structure but nevertheless show also greatest difference in this C-terminal four-helix bundle of Vh2 as well as relative movements of the Vt2 domain (Figure 5B).

### 2.7. The Metavinculin Vh2 C-Terminal Subdomain Constellations

The C-terminal four-helix bundle of the Vh2 domain is in proximity with the N-terminal four-helix bundle of Vh1 and with MVt. In the H1’-kinked metavinculin structure, the distance of the C-terminal Vh2 subdomain to either the N-terminal subdomain of Vh1 or the MVt domain are 9.4 Å (as measured between the Cα positions of residues 73 and 396) and about 13.5 Å (as measured between the Cα positions of residues 468 and 1086). These values are 9.8 Å and 10.8 Å in the H1’-parallel metavinculin structure. Subunit A in the crystal has its Vh2 C-terminal subdomain much more solvent exposed, as these distances are almost doubled (15.7 and 19 Å, respectively). Subunit B in the crystal has values closer to the cryogenic electron microscopy structures (8.8 and 15.3 Å). Thus, the C-terminal Vh2 four-helix bundle seems to be sampling a large conformational space.

### 2.8. The Metavinculin Vt2–MVt Domain Constellations

In addition to the interaction of the N-terminal four-helix bundle of Vh1 (residues 1–129) with the MVt tail domain (residues 946–1132), metavinculin seems to be held in its closed conformer by the intramolecular interaction of the Vt2 domain with the MVt domain (Figure 5C). While there are relative domains movements, all conformations seem to have interactions conserved that keep the Vt2-MVt lock. For example, Vt2 residue Asn-773 is in electrostatic interaction with MVt residue Asp-1042 as well as the Glu-775 interaction with Arg-1046.

Subunit B in the crystal is the only polypeptide chain that has residues visible in the electron density map beyond the Vt2 four-helix bundle domain (through residues 859). In subunit B, Vt2 residue Glu-839 binds MVt residue Arg-1117, thereby stabilizing this loop which is disordered in subunit A in the crystal or in the cryogenic electron microscopy structures. Collectively, interactions between the Vt2 and MVt domains seem an additional conserved hook to keep the metavinculin structures in their auto-inhibited conformations.

### 2.9. The Metavinculin C-Terminal F-Actin Binding Domain MVt

Subunit B in our crystal structures is the most ordered polypeptide chain with electron density only missing for residues 860–946 for the deletion mutant Δ954 or residues 857–949 in our wild-type structure. Our 2.2 Å isolated MVt crystal structure (Protein Data Bank entry 3myi) comprises residues 961–1116 and 1120–1129, which superimposes with H1’-kinked (or H1’-parallel) metavinculin with root mean squares deviations of 1.5 Å (or 1.3 Å) for 924 (or 950) atoms. MVt superposes best with subunit A of our full-length crystal structures (0.4 Å for 927 atoms). These values are 0.5 Å for 915 atoms superimposed onto subunit B.

In our cryogenic electron microscopy structure, we did not observe a definitive density for the C-terminus (residues 1120 to 1134). This suggests that this region is flexible and in various conformations. Recently, the cryogenic electron microscopy structure to 2.9 Å resolution was reported of MVt residues 879–1134 bound to F-actin, but the N-terminus residues 879–980 that include the first α-helix (that we originally named H1’ to emphasize its distinction from the equivalent α-helix H1 of the vinculin isoform) of this five-helix bundle were disordered and could not be modeled [70]. This is consistent with an earlier 8.2 Å reconstruction of MVt (residues 858–1129) that also showed that α-helix H1’ was displaced from the five-helix bundle domain [22]. The coiled coil (residues 949–960) preceding the α-helix H1’ could not be built in our cryogenic electron microscopy structures with residue 840 through 960 missing.

## 3. Discussion

Metavinculin shares functional as well as structural similarities with its splice variant isoform, vinculin, which has been shown to be a very dynamic molecule involved in focal adhesion as well as adherent junctions. The crystal structures of full-length vinculin and metavinculin display a snapshot of the overall conformation of the molecule without their full dynamics. Our 4.15–4.5 Å cryogenic electron microscopy metavinculin conformers uncover the overall dynamics of human metavinculin.

Metavinculin is on the small (about 125 kDa) side for cryogenic electron microscopy but turned out a great tool to elucidate the structural dynamics for smaller proteins. To better resolve and visualize the molecular motions of metavinculin in detail, three-dimensional variability analyses [71] were carried out with the default option of three modes on the entire stack of 1,303,504 particles that was subjected to homogeneous refinement. Out of the three modes computed by three-dimensional variability analyses, one mode provided a glimpse of discrete conformational states presented by metavinculin in solution (Figure 6). However, the other two modes provided subtle twists and lateral movement of the Vh2 domain, although they were not clearly discernable due to the moderate resolution of the particles. Accordingly, the particles in solution exhibit the whole spectrum of conformational flexibilities seen from the heterogeneous refinement. The concerted action of all domains leads to a primed interface state from the 4.15 Å conformer to the 4.5 Å conformer with a concomitant release of α-helix H1’ of the MVt domain.

The presentation of Vt as five-helix bundle as a self-contained conformation on its own (PDB entries 1st6 and 1tr2) [17,18] or in complex with Vh1 (PDB entry 1rke) [72] raises an intriguing question when the release of α-helix H1’, or α-helix H1 in vinculin, happens in cells. Unfolding of the vinculin tail domain upon binding to filamentous actin or acidic phospholipids has been suggested before [15], resulting in the release of the coiled coil region preceding α-helix H1’ (or α-helix H1 in vinculin). The recent electron microscopy structures of the F-actin binding domains of vinculin, metavinculin, or αE-catenin have their first α-helices released from their five-helix bundles as a consequence of binding to F-actin as documented by the isolated F-actin binding domains bound to the actin filament [70,73]. Our cryogenic electron microscopy structure of human full-length metavinculin provides a first glimpse of the α-helix H1’ being released even within the auto-inhibited state as a part of its conformational flexibility. Despite the moderate resolution, we were able to model the partially dissociated α-helix H1’ in the 4.5 Å structure, suggesting that this α-helix is loosely held within the five-helix bundle. Furthermore, α-helix H1’ exhibits a buried surface area of about 720 Å^2^, which is much lower than those exhibited by the rest of the individual α-helices (about 1200 to 1500 Å^2^) as observed in the 4.15 and 4.27 Å resolution conformations, respectively. Whether this α-helix H1’ dissociates during the release of the metavinculin auto-inhibited state is an interesting question for future studies. Such mechanism might allow the tail domain to interact with other protein, such as F-actin, with enhanced affinity.

While the affinities of the vinculin isoforms for binding to F-actin are similar (0.6 ± 0.2 μM for MVt and 0.5 ± 0.1 μM for Vt) [21], metavinculin has been shown to behave differently in F-actin binding and bundling activities in comparison to vinculin [16,23]. The release of α-helix H1’ as part of the activation mechanism could lead to the rearrangement of α-helices H1’ and H1 into microdomains, as suggested earlier [23]. However, our observed pre-release of α-helix H1 suggests that this probably occurs prior to binding to F-actin rather than as a consequence of the interaction leading to the bundling difference observed between the isoforms. The rearrangement of the N-terminal α-helix has been proposed to be a general mechanism for F-actin binding to metavinculin, vinculin, and α-catenin family of proteins [70]. It has also been observed that the C-terminal region of the five-helix bundle of the F-actin binding tail domain is flexible. We also see both these changes in our 4.5 Å H1’-kinked metavinculin conformer as a consequence of conformational flexibility within the auto-inhibited form, suggesting again that the activation process might start with a primed state for various protein interactions (Figure 7).

Our cryogenic electron microscopy structure of human metavinculin and our three-dimensional variability analyses suggest that the overall stability of the auto-inhibited state might be contributed by nearly all of its individual domain components. It is possible that the motifs and structural scaffolds of the protein are involved in contributing toward either structural integrity or functional flexibility or both. The various helix bundle domains seem to provide scaffold stability to the structure of these adaptor proteins. The conformational variability observed in our cryogenic electron microscopy metavinculin structure is in agreement with published ion mobility-mass spectrometry experiments that showed that metavinculin occupies a larger conformational range compared to vinculin [74]. In solution, vinculin was found to be entirely in its closed conformer, while metavinculin displayed additional conformers that included extended or unfolded states.

Since vinculin is a ubiquitous protein involved in cellular structural integrity through focal adhesion and adherens junctions, it is important to be structurally primed to interact with various components within the cell, including phospholipids and other cell adhesion molecules. The conformational landscape that vinculin and metavinculin exhibit seems to provide this structural priming that could get the proteins ready to adapt and engage in cell adhesion function at the very onset of signaling cues such as force or other signaling molecules (such as phosphatidylinositol 4,5-bisphosphate, PIP_2_).

## 4. Materials and Methods

### 4.1. Expression and Purification of Metavinculin

Metavinculin expression and purification were carried as described earlier [16] with slight modifications. Briefly, *Escherichia coli* BL21(DE3) cells expressing metavinculin were lysed by sonication in 20 mM Tris-HCl pH 8, 400 mM NaCl, and 20 mM imidazole containing the ethylenediaminetetraacetic acid-free protease inhibitor cocktail (from Millipore Sigma, Burlington, MA, USA) and clarified by ultra-centrifugation (100,000× *g* for 45 min at 4 °C). The supernatant was subjected to HisTrap Ni-Sepharose affinity column (Cytiva Lifesciences) pre-equilibrated with 20 mM Tris-HCl pH 8 and 150 mM NaCl and eluted with a 500 mM imidazole gradient. The eluate was concentrated and loaded onto a size exclusion chromatography column (Superdex200) that was pre-equilibrated with 20 mM Tris-HCl pH 8.0, 150 mM NaCl, and 0.2 mM tris (2-carboxyethyl) phosphine. The peak fractions were pooled and concentrated. Aliquots were flash frozen in liquid nitrogen and stored at −80 °C.

### 4.2. Metavinculin Cryogenic Electron Microscopy Data Collection

We applied 3 μL of purified human metavinculin at a concentration of 0.5 mg/mL onto a glow discharged 300 mesh Quantifoil (Electron Microscopy Sciences, Hatfield, PA, USA) 1.3/1.2 copper grids maintained at 4 °C and 95% humidity using a Leica GP2 (Leica Microsystems, Buffalo Grove, IL, USA) plunge freezer. The grids were blotted for 4 s and immediately plunge frozen in liquid ethane cryocooled by liquid nitrogen and maintained at −183 °C. The frozen grid was imaged on an EF-Krios (Thermo Fisher Scientific, Waltham, MA, USA) operated at 300 kV with a Gatan K3 imaging system and an energy filter with a slit width of 20 eV, and the data was collected at the National Center for CryoEM Access and Training facility situated at the New York Structural Biology Center, New York, NY, USA. Images were acquired using Leginon [75] at a nominal magnification of 105,000X and a dose rate of 42.45 e^−^/Å^2^/s with a total exposure of 1.6 s containing 40 frames for an accumulated dose of 67.92 e^−^/Å^2^. A total of 2,998 images were collected at a nominal defocus of 1.4 μm. The metavinculin cryogenic electron microscopy data were processed using a calibrated pixel size of 0.4124 Å with 2x binning to a final pixel size of 0.8247 Å.

### 4.3. Metavinculin Cryogenic Electron Microscopy Data Processing

Frame alignment and beam induced motion correction of the raw movies were carried out using MotionCor2 [76]. This step as well as all further processing were carried out in cryoSPARC 2.15 [69]. CTFFIND4 [77] was used to determine defocus values of the micrographs. Images that were having large astigmatism, drift, and low contrast transfer function fit resolution were discarded through manual curation. Initially, a few hundred particles were picked from a few random micrographs and used for template-based picking as implemented in cryoSPARC, resulting in 2,622,418 particles from 2842 images. Then, these particles were extracted using 256-pixel box size, binned by 4 (3.3 Å/pixel), and applied for two-dimensional classification. Most of the particles that were segregated as featureless or excessive noise containing classes were excluded through two rounds of two-dimensional classification resulting in 199 good classes, as being identified with discernable helical bundle features, and 1,469,121 total particles. Then, these particles were re-extracted in the original 256-pixel box size and subjected to initial *ab initio* three-dimensional reconstruction in cryoSPARC. Additional, per-particle defocus and contrast transfer function-fit resolution-based curation resulted in 1,303,504 particles that were used for homogeneous refinement. The resulting initial *ab initio* model was further subjected to heterogeneous refinement. The particles that were clustered in three groups were further subjected to the final round of homogeneous refinement. The local resolution of the individual maps was calculated using the BlocRes [78] as implemented in cryoSPARC. Evaluation of the heterogeneity of the particles were analyzed with three-dimensional variability analyses and depicted using the three-dimensional variability display as implemented in cryoSPARC. Model building was carried out using our human metavinculin crystal structure [16] as the starting model, which was docked into the electron microscopy map using coot 0.9 [79]. Initial real-space refinement of the whole chain was carried out in coot, and after iterative model building, the final model was refined using Phenix by real-space-refinement [80]. The quality of the final model was assessed by Molprobity [81].

## Figures and Tables

**Figure 1 ijms-22-00645-f001:**
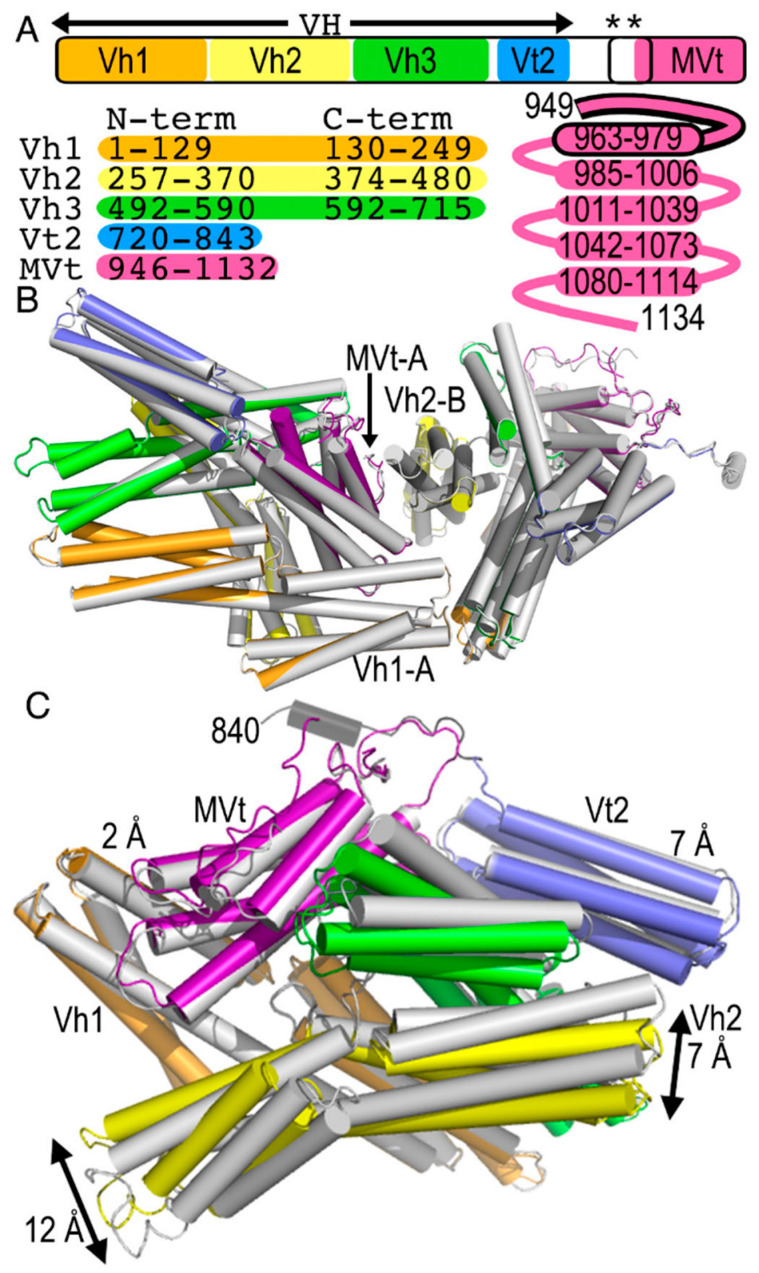
Metavinculin structure. (**A**) Top, Metavinculin domain structure. Metavinculin is organized in four domains colored spectrally (Vh1, orange; Vh2, yellow; Vh3, green; Vt2, blue) that make up the vinculin head (VH) domain that is connected to the metavinculin tail (MVt, violet) domain. The metavinculin-specific insert between vinculin residues 915 and 916 spanning part of the linker and the MVt domain is boxed and indicated by two asterisks. Left, bottom: Vh1, Vh2, and Vh3 have each two subdomains, and their residue range is indicated below the N-term and C-term labels (for N-terminal and C-terminal subdomain, respectively). Right, bottom: MVt is a five-helix bundle domain (residue range of each α-helix is indicated) with an isoform-specific (black borders) coiled coil and first α-helix H1’. (**B**) Superposition of the patient-derived Δ954 metavinculin polypeptide chains in the asymmetric unit in the crystal (gray) onto the two wild type polypeptide chains in the asymmetric unit (Vh1, orange; Vh2, yellow; Vh3, green; Vt2, blue; MVt, violet). Inter-molecular interactions are indicated. (**C**) Superposition of the two polypeptide chains in the asymmetric unit from the structure of the patient-derived ∆954 metavinculin. Subunit A is shown in gray and subunit B is colored spectrally (Vh1, orange; Vh2, yellow; Vh3, green; Vt2, blue; MVt, violet) and its linker region that could be built only in subunit B is shown in black (residues 840–857). Relative domain differences are indicated.

**Figure 2 ijms-22-00645-f002:**
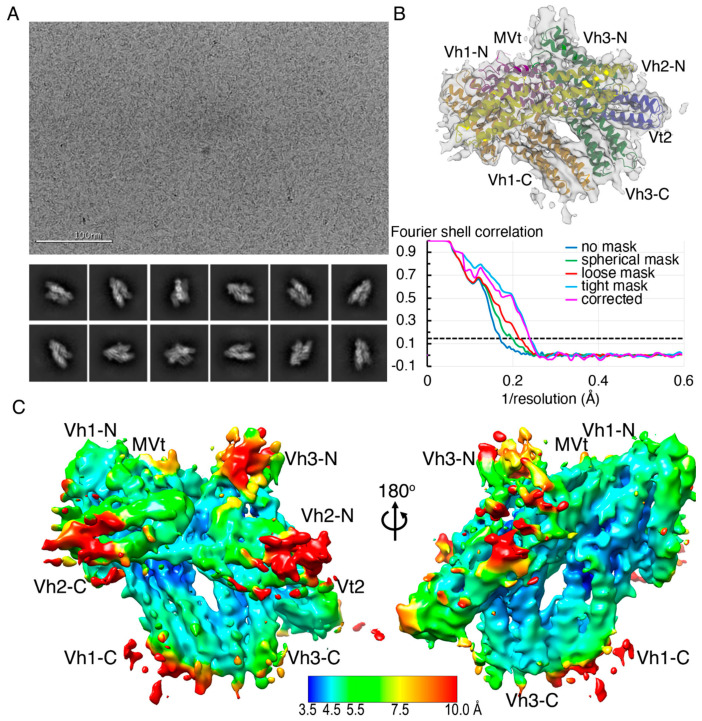
Cryogenic electron microscopy model of full-length metavinculin. (**A**). Top panel, Representative aligned and motion corrected images for metavinculin particles. Bottom panel, Representative two-dimensional classes for metavinculin particles exhibiting visible helix bundle features. (**B**) Top panel, The 4.17 Å metavinculin structure as obtained using all particles is depicted with the docked full-length human metavinculin structure color coded spectrally (Vh1, orange; Vh2, yellow; Vh3, green; Vt2, blue; MVt, violet). Bottom panel, The gold standard Fourier shell correlation curve using a 0.143 Å threshold, as obtained from homogeneous refinement from all particles, is provided for our final 4.17 Å metavinculin structure. (**C**) Local resolution map to show the overall resolution coverage along the entire metavinculin structure. The scale bar is color coded from high (3.5 Å, blue) to low (10 Å, red) resolution.

**Figure 3 ijms-22-00645-f003:**
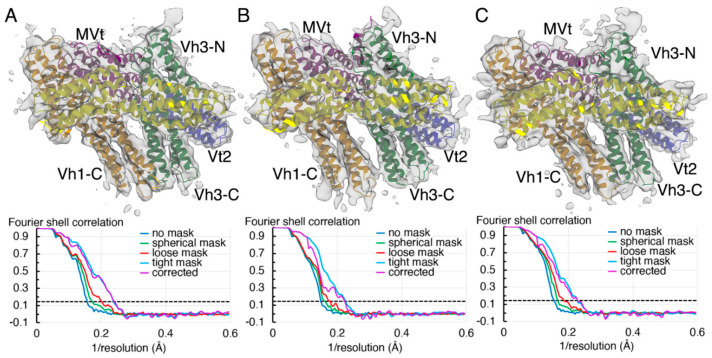
Coordinated metavinculin domain movements confers conformational flexibility. Our metavinculin structure derived from successive heterogeneous and homogeneous refinement elucidates discrete auto-inhibited conformations of metavinculin. The refined metavinculin conformer is colored spectrally (Vh1, orange; Vh2, yellow; Vh3, green; Vt2, blue; MVt, violet). The bottom panels show the gold standard Fourier shell coefficient derived from homogeneous refinement applying a 0.143 Å threshold limit for resolution estimation. (**A**) The 4.15 Å resolution H1’-parallel metavinculin structure is shown with an overlay of the corresponding refined model. (**B**) The 4.5 Å resolution H1’-kinked metavinculin structure exhibiting differently arranged subdomains is shown with an overlay of the corresponding refined model. (**C**) The 4.27 Å resolution H1’-parallel metavinculin structure is shown with an overlay of the corresponding refined model.

**Figure 4 ijms-22-00645-f004:**
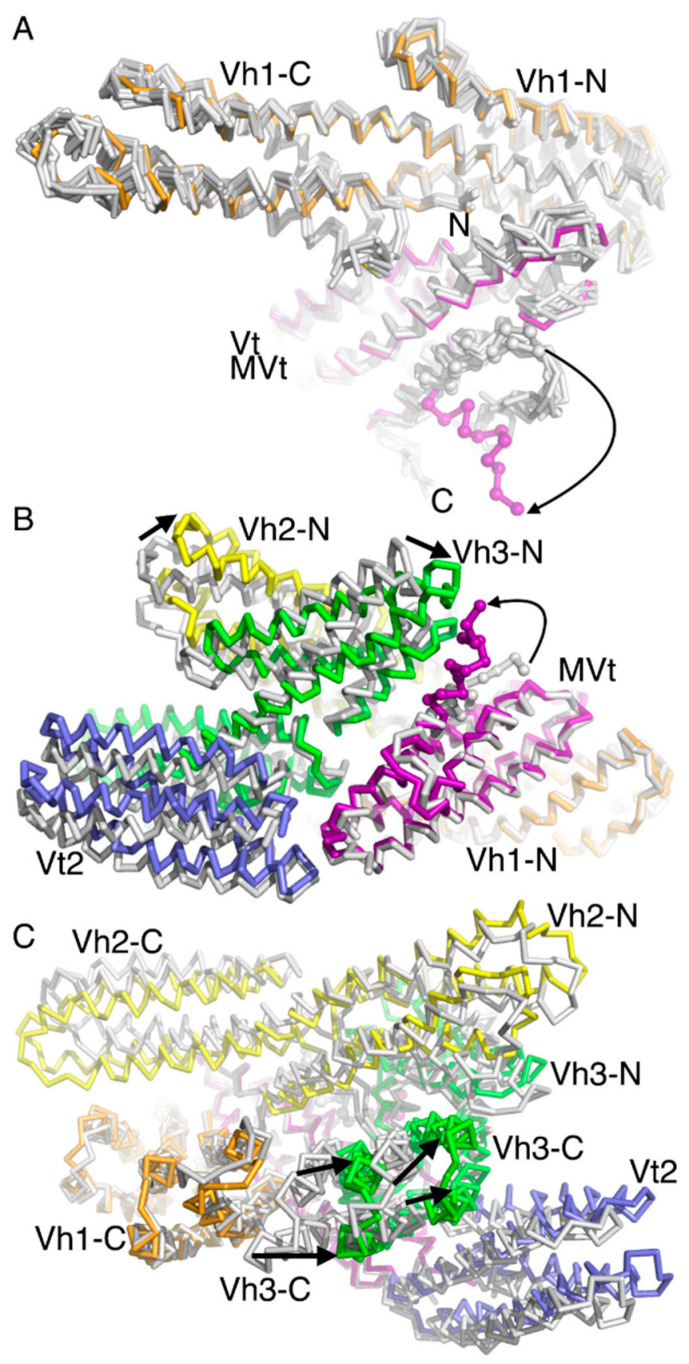
Interdomain distinctions between the H1’-parallel and H1’-kinked metavinculin conformers. (**A**) Superposition of both Vh1 subdomains (Vh1-N for N-terminal and Vh1-C for C-terminal, residues 2–250) and tail MVt (residues 975–1114; Vt in vinculin) domains from our two vinculin subunits in the crystal, our four metavinculin subunits in the crystal, and our three cryogenic electron microscopy structures (all in gray except for the H1’-kinked metavinculin conformer where Vh1 is in orange and MVt in violet) shows that these two domains are almost identical. Cα positions 961–972 of the three cryogenic electron microscopy structures are highlighted as spheres to show the novel kink in the isoform-specific H1’ α-helix (arrow). The termini (residues 1 and 1119) are labeled as N and C, respectively. (**B**) Superposition of the Vh1 (residues 2–250) and MVt (residues 975–1114) domains from our H1’-kinked (with domains colored spectrally: Vh1, orange; Vh2, yellow; Vh3, green; Vt2, blue; MVt, violet) and H1’-parallel (shown in gray) metavinculin cryogenic electron microscopy structures. Cα positions 961–972 of the three cryogenic electron microscopy structures are highlighted as spheres to show the novel kink in the isoform-specific H1’ α-helix (arrow). Several subdomains are labeled, and their relative movements are indicated. (**C**) Superposition of the Vh1 (residues 2–250) and MVt (residues 975–1114) domains from our H1’-kinked (with domains colored spectrally) and H1’-parallel (shown in gray) metavinculin cryogenic electron microscopy structures. Several subdomains are labeled, and the relative movements of the 4 α-helices of the C-terminal Vh3 subdomain are indicated.

**Figure 5 ijms-22-00645-f005:**
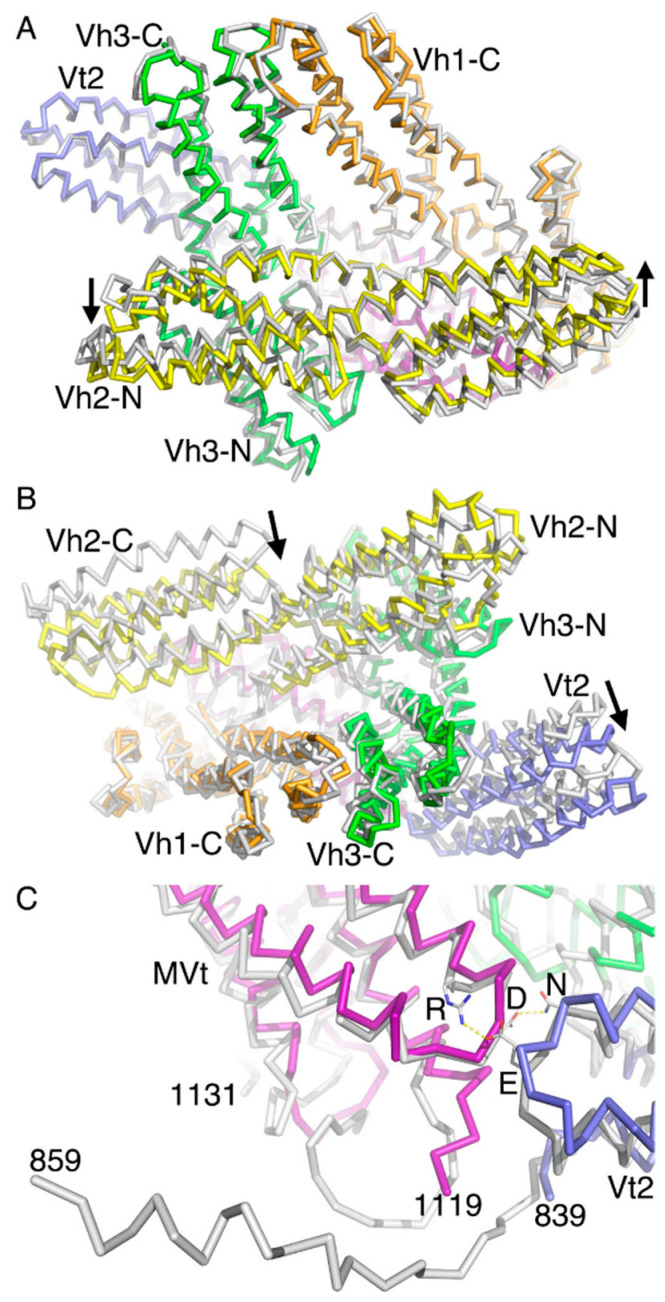
Inter-domain distinctions between the H1’-parallel metavinculin conformers. (**A**) Superposition of both Vh1 subdomains (Vh1-N and Vh1-C, residues 2–250) and the MVt (residues 975–1114) domain from our two H1’-parallel metavinculin cryogenic electron microscopy structures. The 4.15 Å H1’-parallel metavinculin conformer has its domains colored spectrally (Vh1, orange; Vh2, yellow; Vh3, green; Vt2, blue; MVt, violet) and the 4.27 Å H1’-parallel metavinculin conformer is shown in gray. Several subdomains are labeled, and the relative movements of the two Vh2 subdomains are indicated. (**B**) Superposition of the Vh1 (residues 2–250) and MVt (residues 975–1114) domains from our H1’-parallel metavinculin cryogenic electron microscopy (colored spectrally) and crystal (subunit A, shown in gray) structures. Several subdomains are labeled, and relative subdomain movements are indicated. (**C**) Superposition of the Vh1 (residues 2–250) and MVt (residues 975–1114) domains from our H1’-parallel metavinculin cryogenic electron microscopy (colored spectrally) and crystal (subunit B, shown in gray) structures. The crystal structure has residues through 859 for VH and 1131 for MVt, while the cryogenic electron microscopy structure has residues built through 839 for VH and 1119 (labeled) for MVt. Subunit B residues Asn-773, Glu-775, Asp-1042, and Arg-1046 are also labeled by their respective amino acid one letter codes (N, E, D, R).

**Figure 6 ijms-22-00645-f006:**
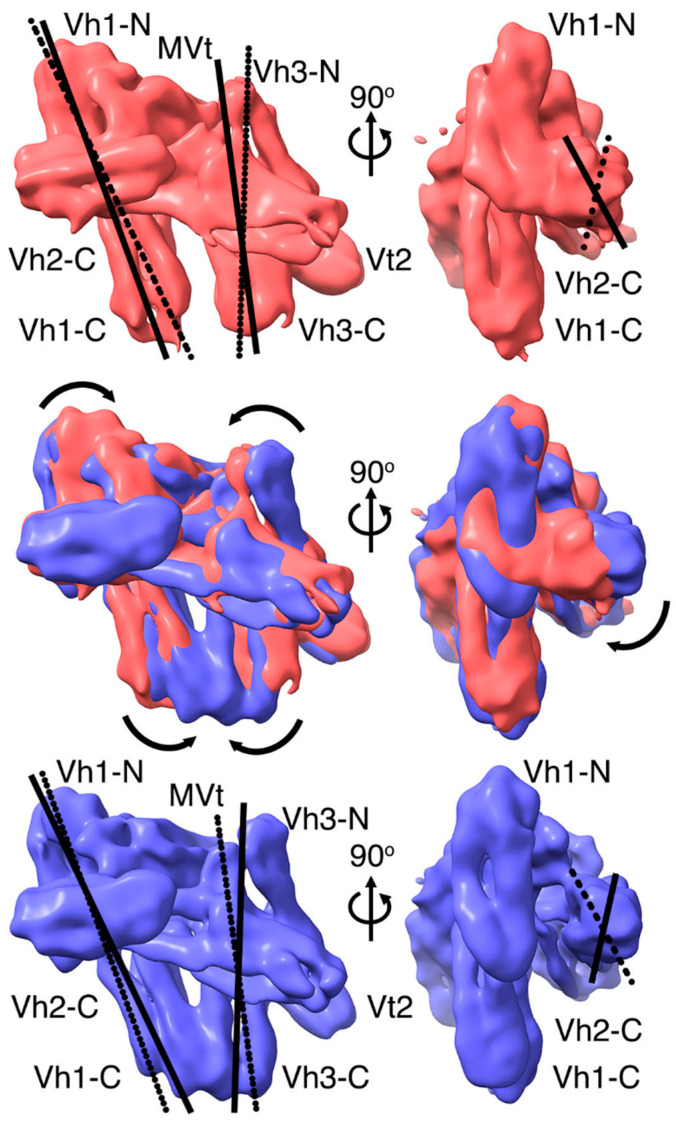
Conformational flexibility as determined by three-dimensional variability analyses. Three-dimensional variability analyses of metavinculin particles of the primed interface in the H1’-kinked state (red) and H1’-parallel state (blue) of metavinculin depicting the overall mobility of the various subdomains. The left panel show the prominent lateral movements of the Vh1 and Vh3 C-terminal four-helix bundles relative to each other. The right panels represent the 90° rotated view and illustrates the lateral movement of the Vh2 C-terminal four-helix bundle toward the Vh1 N-terminal four-helix bundle as a consequence of coordinated movement of rest of the four-helical bundles. The middle panels show the superposed view of the observed states with the top panel showing one of the states and the bottom panel showing the other state in opposite direction. The arrows indicate the direction of movement of individual subdomains, and the solid and dashed lines distinguish the lateral movements corresponding to each other. The various domains positions are labeled for clarity.

**Figure 7 ijms-22-00645-f007:**
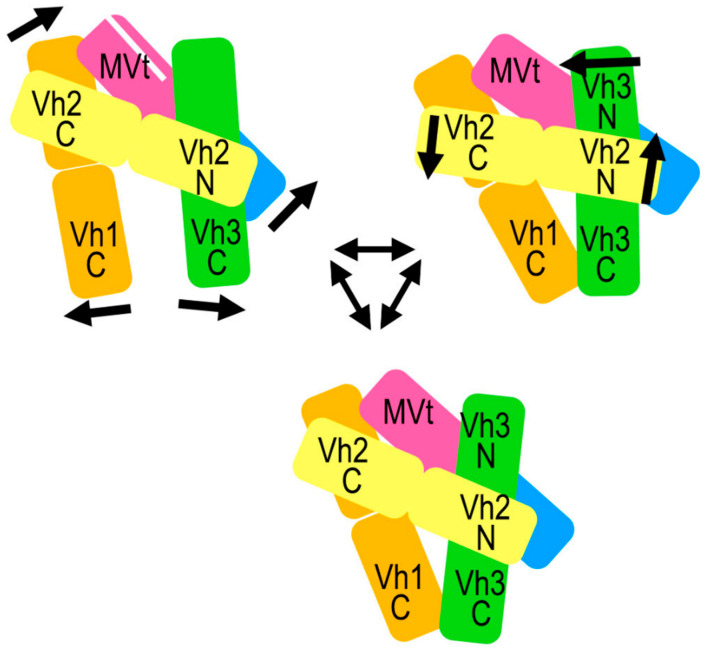
Mechanism of primed interface states exhibited by metavinculin. The cartoon representation of the overall conformation, similar to our crystal structure, is as depicted in the bottom. A slight twist in the four-helix bundles of Vh2 and Vh3 leads to the second H1’-parallel conformation (top right). Either of the H1’-parallel conformations (top right or bottom) transitions to the H1’-kinked conformation accommodating very large movements in the Vh1, Vh3, Vt2, and MVt subdomains with the separation of α-helix H1’ depicted with white separator. (top left). The interplay between these states as seen in our cryogenic electron microscopy structures provides a glimpse of the primed interface state of metavinculin. The subdomains are color coded to represent Vh1 (orange), Vh2 (yellow), Vh3 (green), Vt2 (blue), and MVt (purple). The single arrows indicate dhte domain movements, the double arrows the transitions from one metavinculin conformer to another.

**Table 1 ijms-22-00645-t001:** Metavinculin cryogenic electron microscopy data collection and model statistics.

(**A**) Metavinculin electron microscopy data collection and model reconstruction				
**Conformer**	**Total**	**4.15 Å**	**4.5 Å**	**4.27 Å**
Microscope	EF-Krios	EF-Krios	EF-Krios	EF-Krios
Voltage (kV)	300	300	300	300
Detector	Gatan K3	Gatan K3	Gatan K3	Gatan K3
Magnification (nominal)	105,000	105,000	105,000	105,000
Dose rate (e^−^/Å^2^/sec)	42.45	42.45	42.45	42.45
Frames per exposure	40	40	40	40
Total exposure (e^−^/Å^2^)	67.92	67.92	67.92	67.92
Pixel size (Å/pix)	0.825	0.825	0.825	0.825
Defocus (mm)	1.4	1.4	1.4	1.4
Micrograph collected	3011	3011	3011	3011
Initial particles (no.)	1,303,504	1,303,504	1,303,504	1,303,504
Symmetry	C1	C1	C1	C1
Resolution (Å)	4.17	4.15	4.5	4.27
Particles used (no.)	1,303,504	458,305	413,507	366,802
**(B)** Metavinculin model refinement and statistics of our three distinct conformers				
**Conformer**	**Total**	**4.15 Å**	**4.5 Å**	**4.27 Å**
Atoms	7568	7577	7577	7577
Protein Residues	994	995	995	995
Bonds (root means squares deviations)				
Length (Å)	0.007	0.006	0.006	0.007
Angles (°)	1.016	0.955	1.030	1.032
Molprobity score	2.00	2.12	2.17	2.02
Clash score	12.79	12.91	17.86	14.28
Ramachandran plot				
Favored (%)	94.34	95.66	93.54	94.85
Allowed (%)	5.66	4.34	6.46	5.15
Outliers (%)	0.00	0.00	0.00	0.00
Rotamer outliers (%)	0.25	0.37	0.12	0.12
Cβ outliers (%)	0.00	0.00	0.00	0.00
Mean temperature factors	158.11	217.30	217.93	210.82

## Data Availability

The cryogenic electron microscopy models were deposited with EMDB with accession codes EMD-23029, EMD-23030, EMD-23031, and EMD-23032. The corresponding coordinates were deposited with the Protein Data Bank, accession codes 7ktt, 7ktu, 7ktv, and 7ktw, respectively.

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
