# Peer review of "The Cryogenic Electron Microscopy Structure of the Cell Adhesion Regulator Metavinculin Reveals an Isoform-Specific Kinked Helix in Its Cytoskeleton Binding Domain"

_ijms, 2021, doi:10.3390/ijms22020645_

Round 1
Reviewer 1 Report
Rangarajan and Izard present an interesting study in to the heart-specific vinculin isoform, metavinculin. Metavinculin contains an insertion within the vinculin structure. This leads to a kinked isoform specific alpha-helix in the F-actin binding domain. The authors suggest this modification primes metavinculin to response to mechanical cues.
Overall, I support this study and believe the publication will be of interest to the field. I have the following comments/questions:
- The article needs a careful proof-read throughout.
- The introduction is very long and reads like a review – the authors should make it more concise.
- Line 46 – The authors state that the amino acid sequence of vinculin/metavinculin are “identical”. I understand the message the authors intended but I believe the word “identical” is not correct.
- Fig2- The color code scale should have units added.
- Was the His-tag removed from the constructs? If not, could this impact the structure/conclusions?
- It would be interesting to measure the thermal stability/protein unfolding to compare vinculin and metavinculin. The loosely held alpha helix may lead to an ease of unfolding which would support the structural conclusions.
- Is there a difference in binding affinity between vinculin and metavinculin for F-actin?
Author Response
Reviewer #1
We thank this reviewer for their time in carefully reviewing our manuscript and we are grateful for their helpful suggestions that improved our manuscript significantly.
1. With regards to the comment on careful proof-reading and a more concise introduction, we removed the following 5 paragraphs:
The six amino-terminal 4-helix bundles are paired up into three domains, called Vh1, Vh2, and Vh3, for vinculin head 1 through 3 (17,18). Vh1 is more rigid as its two (N- and C-terminal) 4-helix bundles are connected by one long shared a-helix, whereas in Vh2 and Vh3, there is a short flexible linker of three (in Vh2) or one (in Vh3) amino acid between the two 4-helix bundles. Vh3 is connected to one single 4-helix bundle domain that we called Vt2 for its similarity to Vt and MVt in being a standalone helix bundle compared to the 4-helix bundles of Vh1-Vh3 where two 4-helix bundle subdomains pair up into one domain. Vh1, Vh2, Vh3, and Vt2 are collectively called the vinculin head, VH.
Binding of vinculin and metavinculin in their closed and thus inactive conformations to other proteins is very weak compared to the vinculin and metavinculin binding to such binding partners in their active conformations.
Binding to the plasma membrane or to other large cytoskeletal proteins, such as a-actinin, a-catenin, or talin, that are also made up from 4- and 5-helix bundles, activates vinculin and metavinculin by severing their Vh1 head interaction with their respective Vt or MVt tail domains. The vinculin binding site of these all-helical proteins is an amphipathic a-helix that binds to the second, third, and fourth a-helices of the N-terminal Vh1 4-helix bundle intermolecularly similarly as seen for the intramolecular interaction of the first Vh1 a-helix with these helices. Complex formation results in the first Vh1 a-helix moved to the side and the 4-helix Vh1 bundle forming a 5-helix bundle with the incorporated vinculin binding site. This helix bundle conversion mechanism completely distorts the Vh1 interface with Vt or MVt, thus severing this nanomolar (meta)vinculin head-tail interaction (29-34). Pathogens such as Shigella and Rickettsia use this mechanism by mimicry to exploit this vinculin binding site for entry into the host cell (35-38).
In their open conformation, vinculin and metavinculin bind the actin cytoskeleton readily.
Thus, the function of the six middle 4-helix bundle subdomains (which are the C-terminal 4-helix bundle of Vh1, both 4-helix bundles of Vh2 and Vh3, and the 4-helix Vt2 bundle) seem to provide vinculin and metavinculin the plasticity to connect proteins within the cell junctions and act as a scaffold. This is particularly evident as the complex formed by (meta)vinculin bound to its binding partners can in turn regulate well over one hundred proteins through interactions of the proteins that are activated by (meta)vinculin. The membrane-bound form which also occurs via interactions of the tail domains is additionally functionally and structurally interesting as phospholipid binding causes vinculin and metavinculin oligomerization (64-66).
- We also shortened this sentence:
Vinculin and metavinculin are exclusively helical without any b-sheet which provides these proteins with a large degree of flexibility to easily act as adaptor proteins.
- To now read:
Vinculin and metavinculin are exclusively helical which provides these proteins with a large degree of flexibility to easily act as adaptor proteins.
2. With regards to the comment on our poor choice of the word “identical”, we changed this sentence:
While the amino acid sequence of vinculin and metavinculin are identical, metavinculin has 68 amino acids inserted into the vinculin polypeptide chain between vinculin residues 915 and 916.
- To now read:
Metavinculin has 68 amino acids inserted into the vinculin polypeptide chain between vinculin residues 915 and 916.
3. With regards to the comment about the units, we have now added these to Figure 2C
4. With regards to the question about the Histidine-tag, it was not removed and is at the C-terminus. Since it is disordered and not visible in our structure, we believe it is of no effect.
5. With regards to the suggestions to measure the thermal stability to monitor the protein unfolding to compare the vinculin isoforms, we believe that correlating the local conformational variability by overall unfolding might be difficult. Instead, we now include a discussion on the molecular dynamics study by Benny Geiger’s team that was published in Scientific Reports in 2018 as follows:
The conformational variability observed in our cryogenic electron microscopy metavinculin structure is in agreement with published ion mobility-mass spectrometry experiments that showed that metavinculin occupies a larger conformational range compared to vinculin (74). In solution, vinculin was found to be entirely in its closed conformer, while metavinculin displayed additional conformers that included extended or unfolded states.
6. With regards to the question about affinities, we now provide this information as follows in our revised manuscript:
While the affinities of the vinculin isoforms for binding to F-actin are similar (0.6 ± 0.2 μM for MVt and 0.5 ± 0.1 μM for Vt) (21), metavinculin has been shown to behave differently in F-actin binding and bundling activities in comparison to vinculin (16,23).
Reviewer 2 Report
In the manuscript, the authors capture the structural features of metavinculin, a protein involved in cellular adhesion and a cardiac-cell specific variant of vinculin. They determine both the crystal structure and the cryo-EM structure of the metavinculin protein. In keeping with what is known about the structure and activation of vinculin, the structure of metavinculin appears to indicate the activation processes is conserved between the two proteins. Of note, subtle changes in helices, particularly the kink in the positioning of the first alpha helix of the MVt. These findings are interesting and supported by the evidence presented. A few issues need to be addressed:
Add a figure with the mechanism of action to illustrate the description in the text
If the violet is meant to rep entire MVt, then use different color for first a-helix in Fig 1 bottom right
Describe this patient mutations in more detail- what is the disease, what effect might it have on the process of activation and function described above. In this context, showing the superimposition of vinculin and metavinculin structure will be useful as well as the WT and mutant metavinculin later or in the discussion.
Minor:
Formatting of figure legends- font size and type consistency
Line 66: “Superposition of the two patient derived 954 metavinculin polypeptide chains” seems to imply that there 2 different patient mutations. Reword to clarify that it is 2 chains in the patient-derived protein
Materials and Methods: Specify from which cells the protein was isolated (Bacteria? Human? Mammalian?)
In the discussion suggest potential areas of experimentation to validate the function or mechanism of different conformations observed.
Author Response
Reviewer #2
We thank this reviewer for their time in carefully reviewing our manuscript and we are grateful for their helpful suggestions that improved our manuscript significantly.
1.With regards to the suggestion of adding a mechanistic Figure, we now provide such Figure (new Figure 7).
2. With regards to the violet representing the entire MVt, we now distinguish the metavinculin specific insert by a black border instead. Thank you for pointing that out.
3. With regards to the comment about the patient mutations, our manuscript has been modified according to a 12/17/2020 publication to now read:
Metavinculin point mutations are particular to patients with dilated cardiomyopathy (65) as well as hypertrophic cardiomyopathy (66,67). However, family linkage analyses have not been reported and the number of identified patients with metavinculin mutations seems too small as direct evidence for metavinculin dysfunction causing cardiomyopathies. Loss of metavinculin did not affect the development of the heart and its function (68). Nevertheless, the vinculin isoforms display distinct mechanical properties, and it will be interesting to determine how metavinculin modulates cell adhesion mechanics.
4. With regards to the superposition of metavinculin onto vinculin, we are uncertain which vinculin and metavinculin structures to choose as superimposing all of them would be too crowded to see anything. Please see the superposition (for reviewers only) of the two vinculin subunits (in magenta and yellow) in the crystal onto the two metavinculin subunits (in gray and orange) in the crystal. Adding our 3 cryogenic electron microscopy structures would result in even more crowding.
5. With regards to the inconsistency of the font size and type, this seems to be introduced by the uploading of our WORD document and conversion into the PDF by the journal. We will work with the journal to get this corrected. Many apologies and many thanks for pointing that out.
6. With regards to the wording of Figure legend 1C (somehow the line numbering is also different in our WORD version compared to what the reviewer is seeing who wrote line number 66), this has been reworded, as suggested, to read:
Superposition of the two polypeptide chains in the asymmetric unit from the structure of the patient derived Δ954 metavinculin.
7. With regards to the host cells, we now specify that we used E. coli BL21(DE3) in the methods section.
8. With regards to the potential areas of experimentation to validate the function or mechanism, we addressed this in response to point number 3 where we now state in the manuscript:
it will be interesting to determine how metavinculin modulates cell adhesion mechanics.
